# Molecular Biological and Clinical Understanding of the Pathophysiology and Treatments of Hyperuricemia and Its Association with Metabolic Syndrome, Cardiovascular Diseases and Chronic Kidney Disease

**DOI:** 10.3390/ijms22179221

**Published:** 2021-08-26

**Authors:** Hidekatsu Yanai, Hiroki Adachi, Mariko Hakoshima, Hisayuki Katsuyama

**Affiliations:** Department of Diabetes, Endocrinology and Metabolism, National Center for Global Health and Medicine Kohnodai Hospital, 1-7-1 Kohnodai, Chiba 272-8516, Japan; dadachidm@hospk.ncgm.go.jp (H.A.); d-hakoshima@hospk.ncgm.go.jp (M.H.); d-katsuyama@hospk.ncgm.go.jp (H.K.)

**Keywords:** cardiovascular diseases, chronic kidney disease, hyperuricemia, uricosuric, xanthin oxidase

## Abstract

Uric acid (UA) is synthesized mainly in the liver, intestines, and vascular endothelium as the end product of an exogenous purine from food and endogenously from damaged, dying, and dead cells. The kidney plays a dominant role in UA excretion, and the kidney excretes approximately 70% of daily produced UA; the remaining 30% of UA is excreted from the intestine. When UA production exceeds UA excretion, hyperuricemia occurs. Hyperuricemia is significantly associated with the development and severity of the metabolic syndrome. The increased urate transporter 1 (URAT1) and glucose transporter 9 (GLUT9) expression, and glycolytic disturbances due to insulin resistance may be associated with the development of hyperuricemia in metabolic syndrome. Hyperuricemia was previously thought to be simply the cause of gout and gouty arthritis. Further, the hyperuricemia observed in patients with renal diseases was considered to be caused by UA underexcretion due to renal failure, and was not considered as an aggressive treatment target. The evidences obtained by basic science suggests a pathogenic role of hyperuricemia in the development of chronic kidney disease (CKD) and cardiovascular diseases (CVD), by inducing inflammation, endothelial dysfunction, proliferation of vascular smooth muscle cells, and activation of the renin-angiotensin system. Further, clinical evidences suggest that hyperuricemia is associated with the development of CVD and CKD. Further, accumulated data suggested that the UA-lowering treatments slower the progression of such diseases.

## 1. Introduction

Uric acid (UA) is synthesized mainly in the liver, intestines, and vascular endothelium as the end product of an exogenous purine from food (100–200 mg/day), and endogenously (500–600 mg/day) from damaged, dying, and dead cells, whereby nucleic acids, adenine and guanine, are degraded into UA [1]. There are two types of nucleotide production pathways: a de novo synthesis which newly creates purines and a salvage pathway that reuses purines (Figure 1). Such nucleotides are metabolized to xanthine, which is finally converted to UA by the action of xanthine oxidase (XO). Approximately daily 700 mg of UA is produced by such processes. The kidney plays a dominant role in UA excretion, and the kidney excretes approximately 70% of daily produced UA [2]. The remaining 30% of UA is excreted from the intestine [3]. Usually, the UA pool size of an adult male is about 1200 mg [4]. The UA production is balanced by the excretion of UA into urine (500 mg) and intestine (200 mg). When UA production exceeds UA excretion, hyperuricemia, which has been defined as serum UA concentration > 7.0 mg/dL [5], occurs.

Hyperuricemia is induced by UA over-production due to acquired factors such as high purine diet, fructose ingestion, alcohol intake, myeloproliferative disorders, and also rare genetic causes such as hypoxanthine-guanine phosphor-ribosyl-transferase (HPRT) deficiency and phosphor-ribosyl-pyrophosphate (PPRP) synthetase (PRS) hyperactivity [6]. Renal excretion of UA is the major regulator of serum UA concentration [7], [8]. In humans, reabsorption of UA into the blood plays a crucial role to regulate serum UA. The UA exchange is mediated by various molecules expressed in renal proximal tubule [9]. Such molecules include glucose transporter 9 (GLUT9) [10], urate transporter 1 (URAT1) [11], and human ATP-binding cassette, subfamily G, 2 (ABCG2) [12], organic anion transporter (OAT)1, 3, and 4 [6]. UA enters the cell in exchange for monocarboxylate via apical URAT1 and for dicarboxylate via apical OAT4 [6]. OAT1 and OAT3, on the basolateral membrane of epithelial cells, transport UA from the renal interstitial into renal proximal tubule epithelial cells [13]. Renal UA reabsorption is mainly mediated by URAT1 and GLUT9 [11,14,15]. URAT1 is found in the apical membrane of proximal tubule epithelial cells [6]. Apical GLUT9 plays a significant role in UA reabsorption, the reabsorbed UA exiting the cell through basolateral GLUT9 [6]. In addition, ABCG2 has been identified as a high-capacity UA exporter that mediates renal and/or extra-renal UA excretion [16]. ABCG2 is now known to be involved as well in UA excretion into the intestine [16].

Hyperuricemia has been previously classified into the “UA overproduction type”, “UA underexcretion type” and “combined type”, and approximately 10%, 60%, and 30% of hyperuricemic patients have been classified into each type, respectively [5,17]. Abcg2-knockout mice showed increased serum UA and renal UA excretion, and decreased intestinal UA excretion [18], indicating that a significance of decreased extra-renal UA excretion caused by ABCG2 dysfunction for hyperuricemia. At present, hyperuricemia is classified into “renal UA overload type” (“UA extra-renal underexcretion type” and “UA overproduction type”), “UA underexcretion type” and “combined type” [5].

Hyperuricemia was previously thought to be simply the cause of gout and gouty arthritis. Further, the hyperuricemia observed in patients with renal diseases was considered to be caused by UA underexcretion due to renal failure since the kidney excretes approximately 70% of daily produced UA [2]. Therefore, hyperuricemia was not considered as an aggressive treatment target. Here, we show that hyperuricemia itself is associated with the development and severity of metabolic syndrome, cardiovascular diseases (CVD) and chronic kidney disease (CKD), and possible beneficial effects of the treatment for hyperuricemia on CVD and CKD.

## 2. Hyperuricemia and Metabolic Syndrome

### 2.1. Hyperuricemia and the Risk and Severity of Metabolic Syndrome

Hyperuricemia is significantly associated with the development and severity of metabolic syndrome. Meta-analysis showed that higher serum UA levels led to an increased risk of metabolic syndrome regardless of the study characteristics, and were consistent with a linear dose-response relationship [19]. Choi HK, et al. determined the prevalence of the metabolic syndrome at different serum UA levels by using data from 8669 participants aged 20 years and more in The Third National Health and Nutrition Examination Survey (1988–1994) [20]. They found that the prevalence of metabolic syndrome increases substantially with increasing serum UA levels. A nested case-cohort study of 431 patients with 220 cases demonstrating new vascular events during follow-up, originating from the Second Manifestations of Arterial Disease study, showed that serum UA levels were higher in patients with the metabolic syndrome than in patients without. Serum UA concentrations increased with the number of components of the metabolic syndrome adjusted for age, sex, creatinine clearance, and alcohol and diuretic use [21].

Takahashi, S, et al. evaluated the effect of accumulation of intraabdominal visceral fat on the metabolism of UA in 50 healthy male subjects [22]. Multivariate analyses showed that the size of the visceral fat area was the strongest contributor to an elevated serum UA and a decrease in UA clearance. Magnitude of insulin resistance and serum UA concentration were significantly related (r = 0.69; *p* < 0.001), and insulin resistance was also inversely related to urinary UA clearance (r = −0.49; *p* < 0.002), and urinary UA clearance was inversely related to serum UA concentration (r = −0.61; *p* < 0.001) [23]. Insulin resistance due to visceral fat accumulation may increase serum UA by decreasing renal UA clearance in patients with the metabolic syndrome.

### 2.2. Hyperuricemia and the Components of Metabolic Syndrome

#### 2.2.1. Hypertension

The meta-analysis including 25 studies assessing the association between UA and incident hypertension showed that hyperuricemia was associated with a higher risk of incident hypertension, regardless of whether the effect size was adjusted or not [24]. Adjusted relative risk (RR) was 1.15 [95%confidence interval (CI), 1.06 to 1.26] for a 1 mg/dL increase of serum UA. Another meta-analysis including a total of 18 prospective cohort studies showed that hyperuricemia was associated with an increased risk for incident hypertension (adjusted RR, 1.41; 95%CI, 1.23 to 1.58) [25]. For a 1 mg/dL increase in UA level, the pooled RR for incident hypertension after adjusting for potential confounding was 1.13 (95%CI, 1.06 to 1.20).

#### 2.2.2. Diabetes

A meta-analysis of 12 cohort studies showed that serum UA levels were positively associated with incidence of impaired fasting glucose and type 2 diabetes [26]. Another meta-analysis also showed that serum UA level is positively associated with the development of type 2 diabetes regardless of various study characteristics [27]. In the meta-analysis including eight prospective cohort studies, the combined RR of developing type 2 diabetes for the highest category of serum UA level compared with the lowest was 1.56 (95%CI, 1.39 to 1.76) [28]. Dose-response analysis showed the risk of type 2 diabetes was increased by 6% per 1 mg/dL increment in serum UA level.

#### 2.2.3. Dyslipidemia

The meta-analysis covering 17 studies showed that hyperuricemia increased the likelihood of dyslipidemia and the pooled odds ratio (OR) for the highest UA level vs the lowest UA level was 1.84 (95%CI, 1.49 to 2.28) [29]. In the study aimed to systematically review the association between serum UA concentration and components of pediatric metabolic syndrome, the pooled correlations of UA with triglyceride (r = 0.23, 95%CI, 0.19 to 0.38) and high-density lipoprotein (HDL) (r = −0.28, 95% CI, −0.37 to −0.20) were statistically significant [30].

### 2.3. Possible Molecular Mechanisms for the Development of Hyperuricemia in Metabolic Syndrome

Possible molecular mechanisms for the development of hyperuricemia in the metabolic syndrome were shown in Figure 2. Over-intake of purine may be involved in the development of hyperuricemia in metabolic syndrome, because overeating and lack of physical activity are the main causes of metabolic syndrome. The increased protein level of URAT1 was observed in obesity/metabolic syndrome model mice [31]. Upon high-purine load, insulin resistance enhances UA reabsorption as manifested by up-regulated URAT1 expression and reduces UA excretion in the Otsuka-Long-Evans-Tokushima Fatty rats [32].

Glycolytic disturbances were observed in insulin-resistant and hyperuricemic states [33]. Diversion of glycolytic intermediates toward ribose-5-phosphate (R-5-P), PPRP, and UA will follow if there is diminished activity of glyceraldehyde-3-phosphate dehydrogenase (GA3PDH), which is regulated by insulin. Intrinsic defects in GA3PDH and a loss of its responsiveness to insulin can explain the association between insulin resistance and hyperuricemia.

The meta-analysis of 29 diet-intervention trials showed that short-term fructose consumption promotes the development of hepatic insulin resistance in non-diabetic adults [34]. Another meta-analysis including 3102 articles suggested that fructose consumption from industrialized foods has significant effects on most components of metabolic syndrome [35]. The risk of hyperuricemia and gout was also positively correlated with the intake of fructose (OR, 2.14; 95%CI, 1.65 to 2.78) [36]. In the meta-analysis of prospective cohort studies, fructose consumption was associated with an increase in the risk of gout (RR, 1.62; 95%CI, 1.28 to 2.03; *p* < 0.0001) when comparing the highest and lowest quantiles of fructose consumption [37]. Fructose intake is significantly associated with the development of both hyperuricemia and metabolic syndrome.

In kidney tissue of Sprague–Dawley rats which was induced metabolic syndrome by high-fructose diet, gene expression of GLUT9 was significantly upregulated [38]. Immunohistochemical study showed a significant increase of GLUT9 by more than 3-fold. GLUT9 is a high-capacity urate transporter expressed in the proximal renal tubular cell, which reportedly also transports glucose and fructose [10]. In in vitro assays, the presence of fructose promotes UA transport via this receptor [10]. Variation in GLUT9 gene influences acute serum UA and fractional excretion of UA responses to a fructose load [39]. GLUT9 genotype was suggested to influence the development of gout on exposure to fructose-containing beverages.

Increasing amount of fructose in the diets of rats and humans results in increases in the activity of fructokinase [40]. Accumulation of fructose-1-phosphate causes depletion of ATP and inorganic phosphorus and increases degradation of nucleotides to UA [41].

## 3. Hyperuricemia and CVD

### 3.1. Hyperuricemia and Atherosclerosis

The meta-analysis including 15 studies showed that carotid intima-media thickness (CIMT) in the high UA group was significantly higher than that in the control group [standardized mean difference (SMD), 0.53; 95%CI, 0.38 to 0.68), and the difference was significant (*p* < 0.00001) [42]. Subgroup analysis by disease status illustrated a positive relationship between serum UA levels and CIMT in healthy people and people with diseases. In the meta-analysis and systematic review to explore the modifiable risk factors for carotid atherosclerosis, hyperuricemia was significantly associated with the presence of carotid plaque, and could elevate the risk of atherosclerosis by at least 50% [43].

### 3.2. Hyperuricemia and Coronary Heart Diseases (CHD)

In the meta-analysis including 29 prospective cohort studies, hyperuricemia was associated with increased risk of CHD morbidity (adjusted RR, 1.13; 95%CI, 1.05 to 1.21) and mortality (adjusted RR, 1.27; 95%CI, 1.16 to 1.39) [44]. For each increase of 1 mg/dL in UA level, the pooled multivariate RR of CHD mortality was 1.13 (95%CI, 1.06 to 1.20). Another meta-analysis including 14 studies involving 341, hyperuricemia was associated with an increased risk of CHD mortality (RR, 1.14; 95 %CI, 1.06 to 1.23) [45]. For each increase of 1 mg/dL of serum UA, CHD mortality increased by 20%. According to the gender subgroup analyses, hyperuricemia increased the risk of CHD mortality in women (RR, 1.47; 95 %CI, 1.21 to 1.73) compared to men (RR, 1.10; 95 %CI, 1.00 to 1.19). In a systematic review and meta-analysis using 26 studies, hyperuricemia was associated with an increased risk of CHD incidence (unadjusted RR, 1.34; 95%CI, 1.19 to 1.49) and mortality (unadjusted RR, 1.46; 95%CI, 1.20 to 1.73) [46]. When adjusted for potential confounding, the pooled RR was 1.09 (95%CI, 1.03 to 1.16) for CHD incidence and 1.16 (95%CI, 1.01 to 1.30) for CHD mortality. For each increase of 1 mg/dL in UA level, the pooled multivariate RR for CHD mortality was 1.12 (95%CI, 1.05 to 1.19). Subgroup analyses showed no significant association between hyperuricemia and CHD incidence/mortality in men, but an increased risk for CHD mortality in women (RR, 1.67; 95%CI, 1.30 to 2.04).

### 3.3. Hyperuricemia and Stroke

The meta-analysis of 16 studies showed that hyperuricemia was associated with a significantly higher risk of stroke incidence (RR, 1.41; 95%CI, 1.05 to 1.76) and mortality (RR, 1.36; 95%CI, 1.03 to 1.69) [47]. Subgroup analyses of studies adjusting for known risk factors such as age, hypertension, diabetes mellitus, and cholesterol still showed that hyperuricemia was significantly associated with stroke incidence (RR, 1.47; 95%CI, 1.19 to 1.76) and mortality (RR, 1.26; 95%CI, 1.12 to 1.39). Another meta-analysis including 15 prospective studies indicated that the presence of hyperuricemia was associated with a significantly greater risk of stroke incidence (RR, 1.22; 95%CI, 1.02 to 1.46) and mortality (RR, 1.33; 95%CI, 1.24 to 1.43) [48]. In addition, the pooled estimate of multivariate RRs of stroke incidence and mortality were 1.08 (95%CI: 0.85 to 1.38); 1.26 (95%CI: 1.14 to 1.40) among men and 1.25 (95%CI: 1.04 to 1.46); 1.41 (95%CI: 1.31 to 1.52) among women respectively.

### 3.4. Molecular Mechanisms of Hyperuricemia-Induced Atherogenesis and Thrombosis

Hyperuricemia promotes the occurrence and progression of CVD by regulating molecular signals, such as inflammatory response, oxidative stress, insulin resistance, endoplasmic reticulum stress, and endothelial dysfunction [49]. Hyperuricemia may be also responsible for microvascular damage through stimulation of the renin-angiotensin system (RAS), inhibition of endothelial nitric oxide, and proliferative effects on vascular smooth muscle [50].

Oxidative stress is a well-known component of atherosclerotic pathogenesis, occurring in parallel with activation of pro-inflammatory signaling pathways and expression of cytokines/chemokines [51]. XO uses molecular oxygen as electron acceptor to generate hydrogen peroxide and superoxide anions [52]. XO is normally present in the endothelial cells and in blood, and its levels in atherosclerotic plaques was found to be increased [53]. Several studies highlighted the possible involvement of XO in atherosclerosis development. It was shown that atherogenesis in apo-E knockout mice could be reduced by XO inhibitors [54]. Further, the inhibition of XO reduced the endothelial dysfunction in heavy smokers [55]. XO stimulates the expression of scavenger receptors in macrophages and vascular smooth muscle cells.

Increased UA is also involved in the upregulation of lethal 7-c (let 7-c), which is interconnected with platelets functionality. In a study performed on an animal model of hyperuricemia, increased serum UA generated thrombosis through the activation of myocyte enhancer factor-2C-dependent and nuclear factor-kappa B pathways by let 7-c [56]. In mice, the inhibition of XO led to a decrease in the expression of plasminogen activator inhibitor-1 and tissue factor, suggesting a significant contribution of hyperuricemia to prothrombotic state [57]. Platelet-derived microparticles (PDMPs) are prothrombotic molecules that contribute to thrombus formation. Increased PDMPs may promote hypercoagulable states. The inhibition of XO led to a significant reduction in PDMPs in hyperuricemic patients [58].

## 4. Hyperuricemia and Renal Dysfunction

### 4.1. Hyperuricemia and CKD

In the meta-analysis including 15 cohorts with a total of 99,205 individuals and 3492 incident CKD, the RR of CKD was 1.22 (95%CI, 1.16 to 1.28) per 1 mg/dL serum UA level increment [59]. The observed positive association was more pronounced among group with a mean age < 60 years (RR, 1.26; 95%CI, 1.21 to 1.31). However, no association was observed among studies with a mean age ≥ 60 years (RR, 1.04; 95%CI, 0.96 to 1.13). This mean age-related difference in the association between serum UA levels and CKD was significant (*p* = 0.004). A systematic review and meta-analysis including 13 studies containing 190,718 participants showed a significant positive association between elevated serum UA levels and the new-onset CKD at follow-up (OR, 1.15; 95%CI, 1.05 to 1.25) [60]. Hyperuricemia was found be an independent predictor for the development of newly diagnosed CKD in non-CKD patients (OR, 2.35; 95%CI, 1.59 to 3.46). This association increased with increasing length of follow-up. No significant differences were found for risk estimates of the associations between elevated serum UA levels and developing CKD between males and females.

### 4.2. Molecular Mechanisms of Hyperuricemia-Induced Renal Dysfunction

UA is known to induce hypertension through its effects on endothelial function and impaired production of nitric oxide [61]. Hypertension can be the initial trigger leading to subclinical renal damages [62]. In animal models, hyperuricemia caused hypertension through activation of both vasoactive and inflammatory processes that have multiple effects that include sodium retention and vascular constriction [63]. Histologic analyses showed the lesions that were similar to those seen in hypertension, with the presence of arteriolosclerosis and tubule-interstitial injury [64]. Serum UA levels were significantly correlated with vascular resistance at the afferent, but also efferent, arteriole, suggesting that hyperuricemia may be harmfully associated to glomerular perfusion [65]. Emerging evidence suggests a significant contribution of activation of the RAS by hyperuricemia to the development of CKD [66]. Renal vasoconstriction and reduced renal plasma flow can be induced by activation of RAS. Further, UA may increase oxidative stress, leading to mitochondrial dysfunction, over-secretion of pro-inflammatory cytokines, and proliferation of vascular smooth muscle cells. UA crystals can cause tubular damage through inflammation mediated by direct physical mechanisms.

## 5. Effects of UA-Lowering Treatment (ULT) on CVD and CKD

### 5.1. ULT

The pharmacological action points of ULT were shown in Figure 3. XO is a rate-limiting and catalyzing enzyme of UA formation in purine metabolism, and is involved in reactive oxygen species generation. Allopurinol is classified as purine-like XO inhibitor and febuxostat and topiroxostat are classified as non-purine XO inhibitors. XO inhibitors present antioxidant properties by reducing the production of reactive oxygen species derived from purine metabolism. XO inhibitors still remain the first line of treatment for hyperuricemia.

Uricosuric agents are still second-line or alternative agents, recent guidelines also support the combination of agents such as XO inhibitors and uricosuric agents when monotherapy is not effective [67,68]. Probenecid decreases serum UA by inhibiting URAT1 and GLUT9, and is the prototypical uricosuric agent [69]. Benzbromarone is a potent uricosuric drug that acts by inhibition of URAT 1 and GLUT 9. Benzbromarone was shown to be more potent than probenecid when used as an add-on to daily 300 mg of allopurinol, with 92% of participants reaching a target serum UA of 5 mg/dL [70]. Emerging evidence of the possible role of hyperuricemia in cardiovascular and metabolic comorbidities have led to the development of newer agents. Lesinurad and arhalofenate are the inhibitors of URAT1 and OAT4 [69]. Dotinurad, a selective urate reabsorption inhibitor, is available in Japan. Dotinurad, benzbromarone, lesinurad, and probenecid inhibited URAT1 with IC50 values of 0.0372, 0.190, 30.0, and 165 μM, respectively. Dotinurad weakly inhibited ABCG2, OAT1, and OAT3, with IC50 values of 4.16, 4.08, and 1.32 μM, respectively, indicating higher selectivity for URAT1. [71]. The non-inferiority of dotinurad to febuxostat in terms of serum UA lowering effect was confirmed, and no noteworthy safety concerns arose [72].

### 5.2. Effects of ULT on Atherosclerosis and CVD

In the meta-analysis which assessed the effect of allopurinol on endothelial function, there was a significant increase in the endothelium-dependent vasodilatation with allopurinol treatment [mean difference (MD), 2.69%; 95%CI, 2.49 to 2.89%; *p* < 0.001] [73]. The meta-analysis which studied the effect of allopurinol on flow-mediated dilation (FMD), an index of endothelial function in humans showed that allopurinol therapy significantly improved FMD as compared with control groups [weighted MD (WMD), 1.67%; 95%CI, 0.83 to 2.50%; *p* < 0.001] [74]. In this study, the benefit of allopurinol to FMD seemed to be not related to its UA-lowering action. The meta-analysis including ten eligible randomized controlled trials (RCTs), with showed a significant increase in FMD following allopurinol treatment (WMD, 1.79% 95%CI, 1.01 to 2.56; *p* < 0.001) [75]. No significant association between allopurinol-induced changes in serum UA levels and FMD was found (*p* = 0.253). Further, allopurinol was found to have a statistically significant benefit on endothelial function in patients with chronic heart failure (MD, 0.776; 95%CI, 0.429 to 1.122) and CKD (MD, 0.350; 95%CI, 0.009 to 0.69) but not in type 2 diabetes (MD, 1.331; 95%CI, −0.781 to 3.444) [76].

Zhang, T, et al. performed the meta-analysis to determine if ULT in gout can reduce CV outcomes. Comparing any ULT to placebo (eight studies, *n* = 2221 patients) did not demonstrate a significant difference in non-Anti-Platelet Trialists’ Collaboration CV events (any ULT vs placebo: RR, 1.47; 95%CI, 0.49 to 4.40; *p* = 0.49) or all-cause mortality (any ULT vs placebo: RR, 1.45; 95%CI, 0.35 to 5.77; *p* = 0.60) [77]. In the meta-analysis of major adverse cardiovascular events (MACE), XO inhibitors did not significantly reduce risk of MACE (OR, 0.71; 95%CI, 0.46 to 1.09) and death (OR, 0.89; 95%CI, 0.59 to 1.33), but reduced risk of total CV events (OR, 0.60; 95%CI, 0.44 to 0.82) [78]. There was protection for MACE in patients with previous ischemic events (OR, 0.42; 95%CI, 0.23 to 0.76). Allopurinol protected for myocardial infarction (OR, 0.38; 95%CI, 0.17 to 0.83) and total CV events (OR, 0.48; 95%CI, 0.31 to 0.75). Non-purine-like XO inhibitors did not significantly reduce or increase the risk of adverse CV events. In the meta-analysis to assess the efficacy of allopurinol treatment in reducing the incidence of myocardial infarction following coronary artery bypass grafting, a fixed-effects meta-analysis identified a statistically significant reduced incidence of myocardial infarction (RR 0.21; 95%CI, 0.06 to 0.70; *p* = 0.01) in patients allocated to allopurinol [79]. In seven RCT studies where XO inhibitors were compared with no-treatment or placebo, the results of five low CV risk studies showed that XO inhibitors lowered the risks of both MACE (RR, 0.35; 95%CI, 0.20 to 0.62) and CV events (RR, 0.61; 95%CI, 0.44 to 0.85) [80]. In the meta-analysis to identify studies comparing the efficacy of allopurinol in patients undergoing coronary artery bypass graft (CABG), the pooled OR of periprocedural acute cardiovascular syndrome (OR, 0.25; 95%CI, 0.06 to 0.96; *p* = 0.05) and cardiovascular mortality (OR, 0.22; 95%CI, 0.07 to 0.71, *p* = 0.01) were significantly lower in patients receiving allopurinol during CABG compared to patients in the control group [81]. By contrast, the odds of cardiovascular mortality in the allopurinol group were not significantly different from the control group in patients on long-term allopurinol after acute cardiovascular syndrome or heart failure (OR, 0.33; 95%CI, 0.01 to 8.21; *p* = 0.50) and (OR, 1.12; 95%CI, 0.39 to 3.20; *p* = 0.83), respectively. Similarly, the use of allopurinol did not reduce the odds of recurrent acute cardiovascular syndrome events at two years (OR, 0.32; 95%CI, 0.03 to 3.18; *p* = 0.33).

Febuxostat is approved for the management of hyperuricemia in patients with gout. In November 2017 the FDA released a warning alert on a possible link between febuxostat and CVD reported in a single clinical trial [82]. A meta-analysis was conducted to assess the risk of MACE in patients receiving febuxostat compared to a control group [83]. The pooled RR of MACE for febuxostat was 0.9 (95%CI, 0.6 to 1.5; *p* = 0.96) compared to the control. The RR of CV-related death for febuxostat was 1.29 (95%CI, 1.01 to 1.66; *p* = 0.03). The meta-analysis was done to determine the association of 2 ULT commonly used in clinical practice (febuxostat vs. allopurinol) on MACE [84]. No significant differences were also noted on all-cause mortality (RR, 1.18; 95%CI, 0.99 to 1.41), myocardial infarction (RR, 0.92; 95%CI, 0.72 to 1.18), and stroke (RR, 1.05; 95%CI, 0.77 to 1.43). The meta-analysis of 15 RCTs showed that the use of febuxostat was not associated with statistically significant risk of cardiovascular mortality [risk difference (RD), 0.12%; 95%CI, −0.25 to 0.49%), all-cause mortality (RD, 0.20%; 95%CI, −0.28 to 0.68%), MACE (RD, 0.40%; 95%CI, −0.34 to −1.13%), myocardial infarction (RD, −0.06%; 95%CI, −0.29 to 0.17%), stroke (RD, 0.10%; 95%CI, −0.15 to 0.35%) [85]. The recent meta-analysis of 16 showed that febuxostat had a better safety outcome compared with allopurinol, which was the composite of urgent coronary revascularization (OR, 0.84; 95%CI, 0.77 to 0.90; *p* < 0.0001) and stroke (OR, 0.87; 95%CI, 0.79 to 0.97, *p* = 0.009) [86]. However, the difference was not found in nonfatal myocardial infarction (OR, 0.99; 95%CI, 0.80 to 1.22, *p* = 0.91), cardiovascular related mortality (OR, 0.98; 95%CI, 0.69 to 1.38; *p* = 0.89) and all-cause mortality (OR, 0.93; 95%CI, 0.75 to 1.15, *p* = 0.52). No significant differences in cardiovascular related mortality and all-cause mortality were observed across any subgroup. The cardiovascular safety of febuxostat compared to allopurinol for the treatment of gout remains equivocal.

### 5.3. Effects of ULT on CKD

The meta-analyses which studied whether ULT reduces the progression of CKD showed insufficient data on incidence of end-stage renal disease (ESRD) for analysis and the heterogeneity across included studies, suggesting that adequately powered RCTs are needed to establish whether ULT has beneficial renal effects [87,88]. Another meta-analysis showed that there was no significant difference in change in GFR from baseline between the allopurinol and control arms in five trials, and allopurinol treatment abrogated increases in serum creatinine from baseline in three trials [89]. Only recent meta-analysis showed that XO inhibitors significantly reduced the risk of ESRD compared to the control (three studies; RR, 0.42; 95%CI, 0.22 to 0.80) and also improved eGFR in data pooled from RCTs with long follow-up times (>3 months) (four studies; MD, 6.82 mL/min/1.73 m^2^; 95%CI, 3.50 to 10.15) and high methodological quality (blind design) (three studies; MD, 2.61 mL/min/1.73 m^2^; 95%CI, 0.23 to 4.99) [90]. No definite effects were apparently noticed on serum creatinine, proteinuria and albuminuria in this meta-analysis [91].

Such meta-analyses cannot prove renal protective effect of ULT due to the main limitation by the heterogeneity across included studies, and did not include trials using non-purine XO inhibitors (febuxostat and topiroxostat) and uricosuric agents. We show the renal outcome by ULT in Table 1. In RCTs, allopurinol reduced renal events and improved eGFR [91,92,93]. Two studies showed a beneficial effect of febuxostat on eGFR [94,95], however, one study failed to show such effects [96]. In the study which observed the effect of switching from allopurinol to febuxostat for the treatment of hyperuricemia and renal function in patients with CKD, febuxostat reduced serum UA levels and slowed the progression of renal disease in comparison with allopurinol [97]. Topiroxostat treatment reduced urinary protein excretion, however, did not show an influence on eGFR [98,99,100]. Benzbromarone did not show a significant change of eGFR in 2 studies [101,102]. In the cohort study including 874 CKD patients with hyperuricemia, compared with allopurinol, benzbromarone was associated with a reduced risk of progression to dialysis [103]. Among patients who reached the therapeutic target, those with febuxostat and benzbromarone initiation had a significantly lower risk of ESRD as compared with allopurinol.

## 6. Other Drugs to Lower Serum UA

The UA lowering mechanism of non-UA-lowering drugs were shown in Figure 4. The main pharmacological action, the possible mechanisms to reduce serum UA and studies which showed its UA-lowering effects of non-UA-lowering drugs were shown in Table 2.

### 6.1. Estrogen

Increased levels of serum UA in postmenopausal women are thought to be caused by a change in renal urate elimination associated with the loss of female hormones. In the study to evaluate the regulation of renal urate transporter expression by female hormones using ovariectomized mice with or without hormone replacement, estradiol suppressed the protein levels of URAT1 and GLUT9 [111]. The role of sex hormones in hepatic XO activity was investigated by using rat liver [112]. The hepatic XO activity was decreased about 15% by estradiol. Serum UA in 61 postmenopausal women before and during hormone replacement therapy (HRT) (daily 0.625 mg conjugated equine estrogen combined with 2.5 mg oral medroxyprogesterone) were measured [104]. Twenty-four untreated postmenopausal women were used as controls. None of these women were receiving UA-modifying agents. HRT significantly reduced serum UA concentrations throughout 3–12 months in postmenopausal women with hyperuricemia. On the other hand, serum UA concentrations in the control group showed no significant changes for 12 months.

### 6.2. Losartan

Losartan, an angiotensin II receptor antagonist, inhibits URAT1 and GLUT9 [113]. In the meta-analysis including 31 RCTs, losartan reduced serum UA levels (WMD, −1.57 mg/dL; 95%CI, −1.83 to −1.30) as compared with other antihypertensive agents [105].

### 6.3. Fenofibrate

The effects of fenofibrate, an anti-lipidemic drug, on UA metabolism and URAT1 was investigated in healthy male subjects [114]. Fenofibrate decreased serum UA levels by increasing its urinary excretion, most likely through the inhibition of URAT1 by fenofibric acid, its major metabolite. The meta-analysis including nine studies demonstrated that fenofibrate significantly reduced serum UA levels (WMD, −1.32 mg/dL; 95%CI, −1.61 to −1.03; *p* < 0.001) [106]. Another meta-analysis showed a significant reduction in plasma UA concentrations following fenofibrate therapy [107].

### 6.4. Sodium-Glucose Cotransporter 2 Inhibitors (SGLT2i)

Our previous study showed that SGLT2i, oral anti-diabetic drug, significantly reduced serum UA levels [115]. In the meta-analysis including 31 studies, SGLT2i significantly decreased serum UA levels compared with placebo, canagliflozin (WMD, −37.02 μmol/L; 95%CI, −38.41 to −35.63), dapagliflozin (WMD, −38.05 μmol/L; 95%CI, −44.47 to −31.62), empagliflozin (WMD, −42.07 μmol/L; 95%CI, −46.27 to −37.86) [109]. Another meta-analysis also demonstrated that any of the SGLT2i (empagliflozin, canagliflozin, dapagliflozin, tofogliflozin, luseogliflozin or ipragliflozin) significantly decreased serum UA levels compared with control (WMD, −37.73 μmol/L; 95%CI, −40.51 to −34.95]) [110].

SGLT2i improve insulin resistance by reducing body weight [116]. The increased protein level of URAT1 was observed in obesity/metabolic syndrome [31]. Insulin resistance enhances UA reabsorption by upregulating URAT1 expression and reduces UA excretion [31]. SGLT2i may reduce over-expressed URAT1 due to insulin resistance in patients with type 2 diabetes, by improving insulin resistance. Further, SGLT2i increase renal UA elimination by another mechanism. SGLT2i increase the concentration of glucose in the proximal tubules, and glucose may compete with UA for apical GLUT9, reducing UA reabsorption [108].

## 7. Conclusions

Hyperuricemia is significantly associated with the development and severity of metabolic syndrome. Increased URAT1 and GLUT9 expression and glycolytic disturbances due to insulin resistance may be associated with the development of hyperuricemia in the metabolic syndrome. Emerging evidence suggests a pathogenic role of hyperuricemia in the development of hypertension, CKD, and CVD, by inducing inflammation, endothelial dysfunction, proliferation of vascular smooth muscle cells, and activation of the RAS. Highly evidenced studies show a significant association of hyperuricemia to the development of CKD and CVD. Beneficial effects of XO inhibitors, especially, allopurinol, for CVD and CKD have been reported. Further studies should be performed to elucidate the effects of non-purine XO inhibitors (febuxostat and topiroxostat) and uricosuric agents on the development and progression of CVD and CKD. Estrogen, losartan, fenofibrate and SGLT2i reduce serum UA by modulating urate transporters and XO.

## Figures and Tables

**Figure 1 ijms-22-09221-f001:**
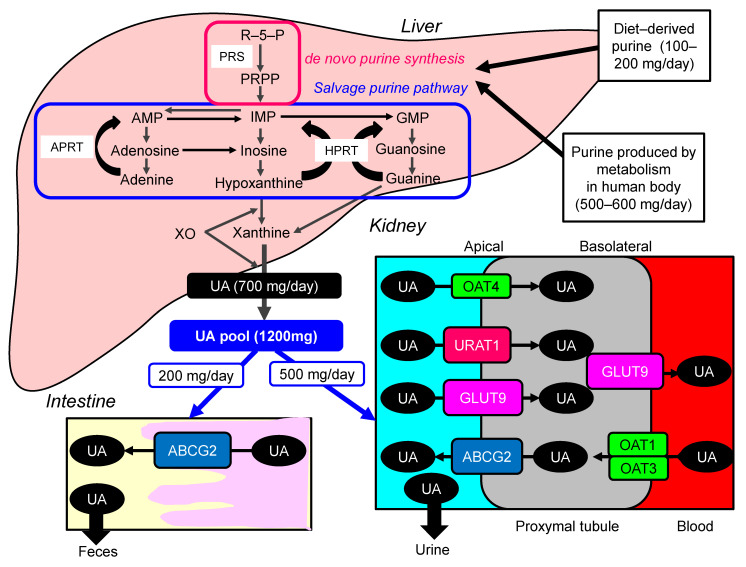
Synthesis and excretion of UA. ABCG2, ATP-binding cassette, subfamily G, 2; AMP, adenosine monophosphate; APRT, adenine phosphoribosyltransferase; GMP, guanine monophosphate; GLUT9, glucose transporter 9; HPRT, hypoxanthine-guanine phosphoribosyltransferase; IMP, inosine monophosphate; OAT, organic anion transporter; PRPP, phosphor-ribosyl-pyrophosphate; PRS, phosphor-ribosyl-pyrophosphate synthetase; R-5-P, ribose-5-phosphate; UA, uric acid; URAT1, urate transporter 1; XO, xanthine oxidase.

**Figure 2 ijms-22-09221-f002:**
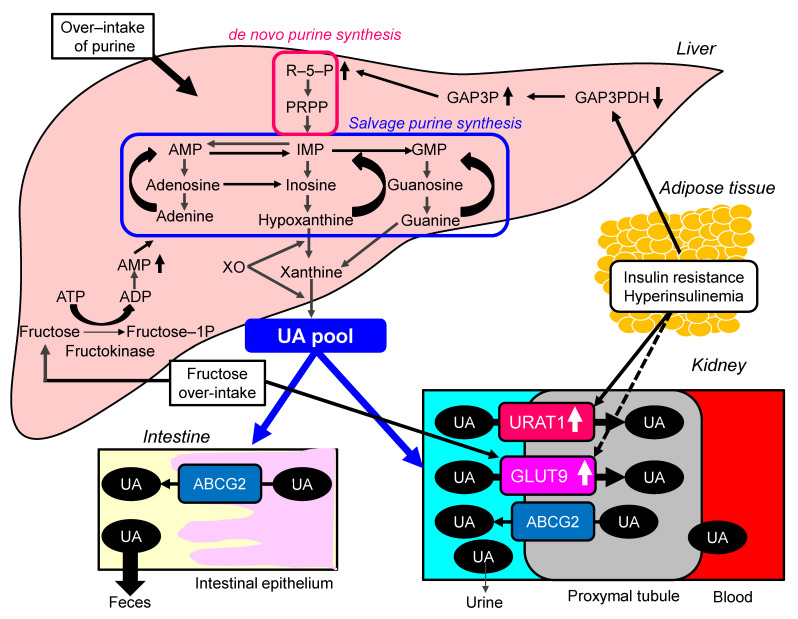
Possible molecular mechanisms for the development of hyperuricemia in the metabolic syndrome. ABCG2, ATP-binding cassette, subfamily G, 2; ADP, adenosine diphosphate; AMP, adenosine monophosphate; ATP, adenosine triphosphate; Fructose-1-P, fructose-1-phosphate; GAP3P, glyceraldehyde-3-phosphate; GA3PDH, glyceraldehyde-3-phosphate dehydrogenase; GMP, guanine monophosphate; GLUT9, glucose transporter 9; IMP, inosine monophosphate; PRPP, phosphor-ribosyl-pyrophosphate; PRS, phosphor-ribosyl-pyrophosphate synthetase; R-5-P, ribose-5-phosphate; UA, uric acid; URAT1, urate transporter 1; XO, xanthine oxidase.

**Figure 3 ijms-22-09221-f003:**
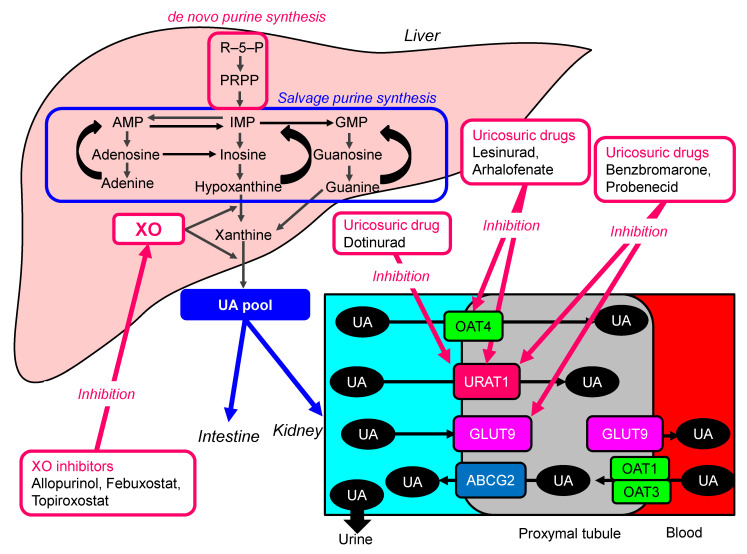
The pharmacological action points of UA lowering treatments. AMP, adenosine monophosphate; GMP, guanine monophosphate; GLUT9, glucose transporter 9; IMP, inosine monophosphate; OAT4, organic anion transporter 4; PRPP, phosphor-ribosyl-pyrophosphate; PRS, phosphor-ribosyl-pyrophosphate synthetase; UA, uric acid; URAT1, urate transporter 1; XO, xanthine oxidase.

**Figure 4 ijms-22-09221-f004:**
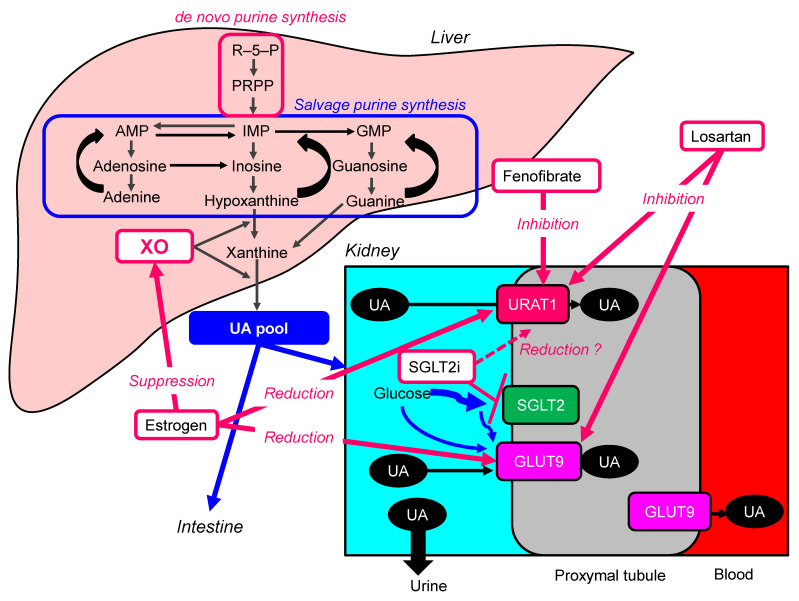
The UA lowering mechanism of non-UA-lowering drugs. AMP, adenosine monophosphate; GMP, guanine monophosphate; GLUT9, glucose transporter 9; IMP, inosine monophosphate; PRPP, phosphor-ribosyl-pyrophosphate; PRS, phosphor-ribosyl-pyrophosphate synthetase; SGLT2i, sodium-glucose cotransporter 2 inhibitors; UA, uric acid; URAT1, urate transporter 1; XO, xanthine oxidase.

**Table 1 ijms-22-09221-t001:** Renal Outcomes by UA Lowering Treatments.

Xanthine Oxidase Inhibitors (The First-Line Treatment for Hyperuricemia)			
Author	Study Design	Subjects Studied	Renal Outcomes
Allopurinol (The most widely used drug approved for the treatment of hyperuricemia)			
Siu, Y.P., et al. [91]	Patients were randomly assigned to treatment with allopurinol, 100 to 300 mg/day, or to continue the usual therapy for 12 months. Study end points included stable kidney function with less than 40% increase in serum creatinine level; impaired renal function with creatinine level increase greater than 40% of baseline value; initiation of dialysis therapy; and death.	54 hyperuricemic patients with CKD	There was a trend toward a lower serum creatinine level in the treatment group compared with controls after 12 months, although it did not reach statistical significance (*p* = 0.08). Less patients (16%) in the allopurinol group reached the combined end points of significant deterioration in renal function and dialysis dependence compared with patients (46.1%) in the control group (*p* = 0.015).
Goicoechea, M., et al. [92]	Patients were randomly assigned to treatment with allopurinol 100 mg/day or to continue the usual therapy. Study end points included renal disease progression; cardiovascular events; and hospitalizations of any causes.	113 patients with eGFR < 60 mL/min.	In the control group, eGFR decreased by 3.3 ± 1.2 mL/min/1.73 m^2^, and in the allopurinol group, eGFR increased by 1.3 ± 1.3 mL/min/1.73 m^2^ after 24 months. Allopurinol treatment reduced risk of CV events in 71% compared with standard therapy.
Goicoechea, M., et al. [93]	Post hoc analysis of a long-term follow-up after completion of the 2-year trial. Intervention is continuation of allopurinol treatment, 100 mg/day, or standard treatment. Study end points included renal event (starting dialysis therapy and/or doubling serum creatinine and/or decrease in eGFR by over 50%) and CV events (myocardial infarction, coronary revascularization or angina pectoris, congestive heart failure, cerebrovascular disease, and peripheral artery disease).	113 participants (57 in the allopurinol group and 56 in the control group) initially followed up for 2 years and 107 participants followed up to 5 additional years.	During the initial and long-term follow-up (median, 84 months), 9 patients in the allopurinol group had a renal event compared with 24 patients in the control group (adjested HR, 0.32; 95%CI, 0.15–0.69; *p* = 0.004). Overall, 16 patients treated with allopurinol experienced CV events compared with 23 in the control group (adjusted HR, 0.43; 95% CI, 0.21–0.88; *p* = 0.02).
Febuxostat (The widely used drug approved for the treatment of hyperuricemia)			
Whelton, A., et al. [94]	A post hoc analysis of the Febuxostat Open-label Clinical trial of Urate-lowering efficacy and Safety study, during which subjects received daily doses of febuxostat (40, 80, or 120 mg) for up to 5 years.	116 hyperuricemic gout subjects	Maintenance or improvement in eGFR was inversely correlated with the reduction in serum UA from baseline. For every 1 mg/dL decrease in serum UA, the model projected an expected improvement in eGFR of 1 mL/min from the untreated value.
Shibagaki, Y., et al. [95]	The safety and efficacy of escalating doses of febuxostat over a 24-week period in patients with CKD stages 3b, 4 and 5 were studied.	70 patients with CKD stages 3b, 4 and 5	Multivariate analysis showed that a greater reduction in serum UA with febuxostat was associated with an increase in eGFR and a tendency toward decreased proteinuria.
Kimura, K., et al. [96]	Participants were randomly assigned in a 1:1 ratio to receive febuxostat or placebo for 108 weeks. The primary end point was the slope of eGFR	467 patients with stage 3 CKD and asymptomatic hyperuricemia at 55 medical institutions in Japan	There was no significant difference in mean eGFR slope between the febuxostat and placebo groups.
Tsuruta, Y. et al. [97]	A 1-year cohort study. In 51 patients, treatment was changed from allopurinol to febuxostat, and the other 22 patients were continued on allopurinol.	73 hyperuricemic patients who had an eGFR below 45 mL/min and were being treated with urate-lowering therapy	The eGFR decreased 27.3 to 25.7 mL/min in the febuxostat group and from 26.1 to 19.9 mL/min in the allopurinol group. The switch from allopurinol to febuxostat was significantly associated with the changes in eGFR according to a multiple regression analysis (β = −0.22145, *p* < 0.05).
Topiroxostat (The drug approved for the treatment of hyperuricemia)			
Hosoya, T., et al. [98]	A 22-week, randomized, multicenter, double-blind study. The enrolled patients were randomly assigned to treatment with topiroxostat 160 mg/day or to the placebo. The endpoints included change in the eGFR, the urinary albumin-to-creatinine ratio.	123 hyperuricemic stage 3 CKD patients with or without gout	After 22 weeks, although the changes in the eGFR was not significant, the percent change in urinary albumin-to-creatinine ratio (−33.0 vs. −6.0 %, *p* = 0.0092) was found to have decreased in the topiroxostat as compared with the placebo.
Horino, T., et al. [99]	Patients were administered 40 mg/day of topiroxostat twice daily. All patients were followed for a year.	30 hyperuricemic patients with CKD (20 male, 10 female)	Topiroxostat treatment resulted in significant reduction in urinary protein excretion (−795.5 mg/gCr) compared with baseline values. However, serum creatinine level, and eGFR did not change significantly.
Katsuyama, H., et al. [100]	Patients who had been continuously prescribed topiroxostat for 3 months or more were retrospectively picked up, and compared serum UA, eGFR and urinary protein before the topiroxostat treatment with the data at 3 and 6 months after the topiroxostat treatment started.	27 hyperuricemic patients	eGFR did not change 3 months after, however, eGFR showed a trend to increase 6 months after. The number of patients who showed positivity for urinary protein significantly decreased at 3 and 6 months after the start of topiroxostat as compared with baseline.
Uricosuric agent (The second-line treatment for hyperuricemia)			
Benzbromarone (The widely used drug approved for the treatment of hyperuricemia)			
Fujimori, S., et al. [101]	Renal function changes over a period of up to 7 years were retrospectively evaluated in patients with CKD associated with hyperuricemia and were receiving monotherapy with benzbromarone	35 patients with CKD (stage 3, 32 patients; stage 4, 2 patients; stage 5, 1 patient) associated with hyperuricemia	No significant changes in eGFR from the baseline value of 46.2 ± 11.5 mL/min/1.73 m^2^ were found after benzbromarone therapy.
Yu, H., et al. [102]	A single-centered, parallel-grouped, RCT. Patients were randomly assigned into benzbromarone and febuxostat treatment group.	66 hyperuricemia participants with eGFR 20–60 mL/min/1.73 m^2^	After 12-month treatment, eGFR did not have significant changes in both groups.
Chou, H.W., et al. [103]	A pharmacoepidemiology cohort study by including patients from Taiwan’s long-term integrated CKD care program to compare the effectiveness among allopurinol, febuxostat and benzbromarone in reducing the risk of progression to dialysis. Patients with hyperuricemia who were newly treated with allopurinol, febuxostat or benzbromarone were included.	874 CKD patients with hyperuricemia	Compared with allopurinol, benzbromarone therapy was associated with a reduced risk of progression to dialysis, the adjusted HR was 0.50 (95%CI, 0.25–0.99). Among patients who reached the therapeutic target, those with febuxostat and benzbromarone initiation had a significantly lower risk of end-stage renal disease.

**Table 2 ijms-22-09221-t002:** Other Drugs to Lower Serum UA.

Drugs	Main Pharmacological Action	Possible Mechanisms to Reduce Serum UA	Studies Which Showed Its UA-Lowering Effects
Estrogen	Female sex hormone	Suppression of the protein levels of URAT1 and GLUT9, and reduction of XO activity	Serum UA in 61 postmenopausal women before and during HRT were measured [104]. HRT significantly reduced serum UA concentrations throughout 3–12 months in postmenopausal women with hyperuricemia. On the other hand, serum UA concentrations in the control group showed no significant changes for 12 months.
Losartan	Antihypertensive drugs, angiotensin II receptor antagonist	Inhibition of URAT1 and GLUT9	In the meta-analysis including 31 RCTs, losartan reduced serum UA levels (−1.57 mg/dL; 95%CI, −1.83 to −1.30) as compared with other antihypertensive agents [105].
Fenofibrate	Anti-lipidemic drug	Inhibition of URAT1	The meta-analysis including 9 studies demonstrated that fenofibrate significantly reduced serum UA levels (−1.32 mg/dL; 95%CI, −1.61 to −1.03; *p* < 0.001) [106]. Another meta-analysis showed a significant reduction in plasma UA concentrations following fenofibrate therapy [107].
Sodium-Glucose Cotransporter 2 Inhibitors (SGLT2i)	Oral anti-diabetic drugs	SGLT2i may reduce over-expressed URAT1 due to insulin resistance in patients with type 2 diabetes, by improving insulin resistance. SGLT2i increase renal UA elimination by another mechanism. SGLT2i increase the concentration of glucose in the proximal tubules, and glucose may compete with UA for apical GLUT9, reducing UA reabsorption [108].	In the meta-analysis including 31 studies, SGLT2i significantly decreased serum UA levels compared with placebo, canagliflozin (−37.02 μmol/L; 95%CI, −38.41 to −35.63), dapagliflozin (−38.05 μmol/L; 95%CI, −44.47 to −31.62), empagliflozin (−42.07 μmol/L; 95%CI, −46.27 to −37.86) [109]. Another meta-analysis also demonstrated that any of the SGLT2i (empagliflozin, canagliflozin, dapagliflozin, tofogliflozin, luseogliflozin or ipragliflozin) significantly decreased serum UA levels compared with control (−37.73 μmol/L; 95%CI, −40.51 to −34.95]) [110].

## Data Availability

Not applicable.

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
