# Peer review of "Molecular Biological and Clinical Understanding of the Pathophysiology and Treatments of Hyperuricemia and Its Association with Metabolic Syndrome, Cardiovascular Diseases and Chronic Kidney Disease"

_ijms, 2021, doi:10.3390/ijms22179221_

Round 1
Reviewer 1 Report
I enjoyed reading the review entitled "Molecular Biological and Clinical Understanding of the Pathophysiology and Treatments of Hyperuricemia and Its Association with Metabolic Syndrome, Cardiovascular Diseases and Chronic Kidney Disease" by Yanai et al.. The manuscript is well-written, well-organised and easy to follow, summarising in a nice way the most important studies on the field.
Author Response
We appreciate the reviewer’s positive comments.
Hidekatsu Yanai, MD, PhD
Reviewer 2 Report
The authors of this review manuscript thoroughly describe the biological mechanisms and clinical studies that demonstrate UA has a causal role in cardiovascular and renal diseases. This review article has important implications for the management of these diseases and will have substantial impact on the field. Several minor suggestions should be addressed prior to publication:
-At the end of the introduction, the authors can put a finer point on the fact that UA was previously thought of as a by-product of cardiorenal disease, whereas it may be an actual cause.
-Are there additional studies or analyses to discuss related to dyslipidemia in section 2.2? This seems to receive much less attention here.
-Addition of clinical use to therapeutics in Table 1 may be helpful, such as frontline vs. second line and whether uses are widespread or approved.
-A table summarizing uses and mechanisms of drugs in Section 6 would also be helpful in describing each possible therapeutic. If this could include clinical trials or uses as well, that would be helpful.
Author Response
Reviewer 2
- According to the comment “At the end of the introduction, the authors can put a finer point on the fact that UA was previously thought of as a by-product of cardiorenal disease, whereas it may be an actual cause.”
We changed to the following sentences.
Further, the hyperuricemia observed in patients with renal diseases was considered to be caused by UA underexcretion due to renal failure because the kidney excretes approximately 70% of daily produced UA [2]. Therefore, hyperuricemia was not considered as an aggressive treatment target. Here, we show that hyperuricemia itself is associated with the development and severity of metabolic syndrome, cardiovascular diseases (CVD) and chronic kidney disease (CKD), and possible beneficial effects of the treatment for hyperuricemia on CVD and CKD.
- According to the comment “Are there additional studies or analyses to discuss related to dyslipidemia in section 2.2? This seems to receive much less attention here.”
We added the following sentence, and cited one more reference.
In the study aimed to systematically review the association between serum UA concentration and components of pediatric metabolic syndrome, the pooled correlations of UA with triglyceride (r = 0.23, 95%CI, 0.19 to 0.38) and high-density lipoprotein (HDL) (r = -0.28, 95% CI, -0.37 to -0.20) were statistically significant [30].
- According to the comment “Addition of clinical use to therapeutics in Table 1 may be helpful, such as frontline vs. second line and whether uses are widespread or approved.”
We changed table 1 to the following table 1.
- According to the comment “A table summarizing uses and mechanisms of drugs in Section 6 would also be helpful in describing each possible therapeutic. If this could include clinical trials or uses as well, that would be helpful.”
We added the following sentences and newly made table 2.
Serum UA in 61 postmenopausal women before and during hormone replacement therapy (HRT) (daily 0·625 mg conjugated equine estrogen combined with 2·5 mg oral medroxyprogesteron) were measured [107]. Twenty-four untreated postmenopausal women were used as controls. None of these women were receiving UA-modifying agents. HRT significantly reduced serum UA concentrations throughout 3–12 months in postmenopausal women with hyperuricemia. On the other hand, serum UA concentrations in the control group showed no significant changes for 12 months.
